# DASHES Protocol: Development and Feasibility Testing of a Tailored Community Programme to Support People in Recovery from Problematic Alcohol and Drug Use to Cut Down or Stop Smoking Using Co-Creation

**DOI:** 10.3390/ijerph192013709

**Published:** 2022-10-21

**Authors:** Fiona Dobbie, Martine Miller, Man Hei Marcus Kam, Aoife McKenna, Claire Glen, Alison McCallum

**Affiliations:** 1Usher Institute, University of Edinburgh, Edinburgh EH8 9AG, UK; 2NHS Lothian, Waverley Gate, 2–4 Waterloo Place, Edinburgh EH1 3EG, UK

**Keywords:** tobacco, addiction, smoking cessation, harm reduction, trauma informed, intervention, development, feasibility

## Abstract

Background: Despite the continued global decline in adult tobacco prevalence, rates continue to be significantly higher in groups with problematic drug or alcohol use (PDA). It is estimated that people with alcohol, drug or mental health problems account for approximately half of all smoking deaths. In the UK, there are free stop smoking services for the general population. However, these services have been criticized as unsuitable for people in recovery from PDA due to their design, time-limited support, strict requirement for smoking abstinence and lack of consideration of harm reduction approaches. This has led to calls for alternative approaches to support this marginalized and underserved group. This research study seeks to respond to this call by co-creating and feasibility testing a tailored, trauma-informed service specifically for people seeking help for PDA, who are not in immediate crisis, and who may also want to reduce or stop their tobacco smoking. Methods: The mixed-method study design has two parts. The development study (part one) will use participatory peer research methods to work with the target client group and key stakeholders involved in service delivery, commissioning, and policy to design the service (intervention). The feasibility study (part two) will test the delivery of the intervention protocol and capture data that will enable the assessment of whether progression to a future pilot randomized control trial is merited. Conclusions: The outcome of this study will be a theoretically informed, co-created intervention with the potential to improve population health by supporting people with problematic drug or alcohol use to cut down or stop tobacco smoking.

## 1. Introduction

The World Health Organization’s recent report on global trends in tobacco prevalence (conducted across 165 countries) shows a continued decline in adult tobacco consumption, from 32.7% in 2001 to 22.3% in 2020 [1]. Despite this decline, smoking prevalence among people with problematic drug or alcohol use (PDA) remains considerably higher. In the UK, for example, it is estimated to be two to four times higher than in the general population (14% in 2020 [2]), with smoking prevalence estimates for people with PDA ranging from 77% to 98% from studies conducted in the USA, Canada, Australia and Switzerland [3].

The associations between smoking, substance use, inequalities, stigmatization and social exclusion reflect global structural inequity and stem from adverse childhood experiences [4]. The intersection of these factors contributes to socioeconomic inequalities, early-onset multimorbidity (including a high prevalence of respiratory disease, cardiovascular disease and mental health problems) and premature mortality [5,6]. It is estimated that people with alcohol, drug or mental health problems account for approximately half of all smoking deaths [7]. Life expectancy of people with PDA is reduced, with underlying causes of death more likely to be tobacco- than alcohol- or drug-related [8,9]. Smoking is, therefore, an important contributory factor to the poor health of current and former PDA users, but it is rarely addressed using theoretically informed and evidence-based approaches. Despite existing research showing the willingness of people with PDA to address their tobacco smoking (especially for those who are further into their recovery journey), there remains an unfounded concern that this may be to the detriment of their recovery [10]. Existing evidence on effective interventions to support this population is limited and predominately from Australia and the USA [11,12].

In the UK, smoking cessation support services exist for the general population. These free services are delivered by the National Health Service (NHS) and are informed by a national set of guidelines and quality standards that are developed and updated by the National Institute for Health and Care Excellence [13,14]. There is no specific guidance on smoking cessation for people in treatment or recovery from PDA [3]. Existing NHS stop smoking services have also been criticized as unsuitable for this client group due to their design, time-limited support and strict requirement for smoking abstinence and without any harm reduction approaches being considered (e.g., cutting down or switching to less harmful products like e-cigarettes). This, combined with a lack of training for smoking cessation staff and the daily stigma the client group face, has resulted in low confidence, poor outcomes and little incentive for staff and clients to participate fully [15].

Previous studies have shown that people who use drugs and alcohol rated positive and inclusive attitudes towards them as key to their successful engagement with addiction treatment services [16]. Stigma reinforces trauma and prevents people from engaging with services and seeking support. Smoking cessation services for this client group must be cognizant of this and be designed in a trauma-informed manner (TI) to support both engagement and retention in services. Trauma-informed service provision recognizes the impact and harm resulting from a user’s often negative or dismissive relationship with services—the stigmatizing and exclusionary nature of services—in line with the key principles of choice, collaboration, trust, empowerment and safety. This means that greater time is often required to build relationships and trust with the service user to enable better engagement in a welcoming and safe environment [17,18,19].

The gap between existing NHS stop smoking services and a trauma-informed model of care has led to calls for alternative approaches to increase engagement and retention. This could be achieved by promoting tobacco harm reduction in order to cut down the number of cigarettes consumed, rather than requiring an initial commitment to complete abstinence [20]. Using a reduction in tobacco consumption approach means that people can feel more in control of their smoking and will becomes less dependent upon nicotine over time, which can make it easier to stop for good. Products like NRT and e-cigarettes contain nicotine (the addictive component of tobacco) but are proven to be less harmful than cigarettes, with some evidence to suggest that e-cigarettes can be more effective than NRT for cessation [21]. Recent studies (using a pilot randomised controlled trial (RCT) and cross-sectional study design) have found NRT and e-cigarettes to be effective in homeless units and residential substance misuse treatment services [12,22]. It is further suggested that the use of these less harmful products may provide an alternative source of pleasure and social interaction that substitutes some of the aspects of smoking that people may miss more than the nicotine itself [23].

In light of these gaps, the overall aim of this research study is to co-create and feasibility test a tailored, trauma-informed service specifically for people seeking help for PDA who are not in immediate crisis but who may also want to reduce or stop their tobacco smoking. It will be situated in day/community services, not residential services, which have been criticized for failing to provide smoking cessation support beyond the residential period [12]. Day/community services also have the advantage of seeing more clients and therefore have the potential for greater reach if future pilot testing is positive.

## 2. Materials and Methods

The DASHES (Drugs and Alcohol Service users Help to Exit Smoking) study will be guided by the new Medical Research Council (MRC) framework for the development and evaluation of complex health behaviour change interventions [24]. The new MRC framework is an update to previous guidance [25]. While the four phases of intervention design are still included (intervention development, feasibility, evaluation and implementation testing), greater emphasis is given to three key areas. First, it notes the importance of clear intervention theory, in terms of the theory that has informed the intervention design but also the intervention theory itself (i.e., how the intervention should work in real life—often illustrated using a logic model). Next is the importance of understanding the intervention context. The new guidance notes that understanding the ways in which context influences intervention outcomes is as important as measuring intervention effectiveness. Finally, it places greater importance on meaningful engagement with the key stakeholders (including people with lived experience, people who will benefit from the intervention and those who will support the successful delivery and future sustainability of the intervention) with co-development of the intervention theory seen as ‘essential.’ These principles are core components of the DASHES study, which will use co-creation approaches to ensure that the intervention is developed in consultation with potential service users; people with lived experience of PDA and those who will plan, commission and deliver services.

Planning and feasibility testing of the intervention will draw on ‘person-based approaches’ (PBA), which are grounded in understanding the needs and experiences of potential service users and other key stakeholders (e.g., service deliverers and commissioners). The PBA offers a systematic approach to intervention design and comprises two elements. The first seeks to generate evidence related to service user experience and the context in which the intervention will be delivered. For DASHES, this will include a mixed-method intervention development study that will include: a rapid literature review, a client survey (delivered by peer interviewers and study researchers), qualitative research and stakeholder workshops. The second component is the development of ‘guiding principles’ that inform intervention theory. These guiding principles are made up of two parts: Part one defines the intervention objectives (i.e., what it is trying to do), and part two describes the specific features of the intervention that address these objectives (i.e., what it will do). For DASHES, this will be presented in a logic model, with a more detailed description outlined using the TIDier checklist [26].

### 2.1. Study Overview

The DASHES study design comprises two work packages delivered over a 24-month period, summarised in Figure 1. Work package 1 will be a development study using participatory peer research methods to work with the target client group and key stakeholders involved in service delivery, commissioning and policy to design the service (intervention).

Work package 2 will be a feasibility study that will test the delivery of the intervention protocol in one Scottish Health Authority and capture data that will enable the assessment of whether progression to a future definitive (RCT) is merited.

### 2.2. Work Package 1: Development Study

This mixed-method development study will consult with a range of stakeholders to inform the development of an intervention that seeks to support people to reduce or stop their tobacco smoking who are currently recovering from PDA. This work package will comprise four strands, with co-creation a key component.

#### 2.2.1. Rapid Review of the Literature

The development study will start with a rapid literature review of the existing published and unpublished ‘grey’ literature to identify best practice approaches to inform service design. For this rapid review, the searching and screening process will be the same as a systematic review (SR), but subsequent stages of a SR, i.e., an exhaustive data extraction of all included studies, will not be included. Instead, we will develop a rapid review protocol informed by good practice [27,28]. This will include details on: (i) the review objectives (which will focus on identifying existing studies/reviews evaluating delivery of smoking cessation support to people in recovery from PDA. These will be assessed for effectiveness, potential delivery models and best practice for client engagement and retention); (ii) databases to be searched (PsycINFO Pubmed, PubMedCentral, Scopus, WoS (Web of Science); (iii) search terms; (iv) search period; (v) search criteria (setting, population, study type); (vi) assessment of data quality (guided by Critical Appraisal Skills Programe tool [29]; (vii) data extraction and synthesis.

#### 2.2.2. Client Survey

To better understand the barriers and facilitators to supporting people with PDA to cut down or stop tobacco smoking, and to identify the key components of this tailored intervention, we will conduct a face-to-face, interviewer-administered, cross-sectional client survey. This will be administered in 10–12 drug and alcohol treatment centres across the central belt of Scotland. The survey will be co-designed and administered by people with lived experience of PDA. We anticipate that the survey will last around 15 min and cover the following areas: current smoking behavior (frequency, peer and environmental influences); previous quit attempts (how many, outcome, methods used to support quit attempt, what helped or hindered their quit attempt); use of and attitudes toward e-cigarettes; awareness, attitude and experience of using NHS stop smoking services); use of and attitudes towards medication, NRT, etc. and thoughts on what kind of smoking cessation service could support them to have a successful quit attempt in the future.

The survey delivery will use participatory peer research methods and will be administered by a third-sector treatment and prevention service (West Lothian Drug and Alcohol Services—WLDAS) with support from the research team. Around six to eight members of the WLDAS Service User Advisory Group (SUAG) will be trained and supported to become peer interviewers.

Data will be collected using convenience sampling and will take place over a three-month period. A hybrid model of data collection will be used that comprises interviewer-administered surveys in the venue; when this is not possible, centre staff will support the self-completion of the survey. We will work with centre staff to position interviewers in the premises at times when we are likely to see the range of clients required to inform the service design. Based on the achieved sample from a previous study using a similar approach [22] and from the scoping study which preceded this study protocol, the target sample is between 120 and 150 respondents. The inclusion criteria will be: be aged 18 or older, be able to give informed consent, have experience of trying to cut down or stop smoking and be at a stable point on their recovery journey (defined as participants who have been accepted into and are participating in a residential or community programme to address their problematic drug or alcohol use. Based on advice from service users and staff, we will support data gathering with language support via a telephone interpretation service.

#### 2.2.3. Qualitative Research

In this qualitative component, we will build on findings from the client survey by adding more depth and understanding around the challenges this client group faces in stopping or cutting down tobacco smoking and what kinds of support they would like. We will achieve this through in-depth consultation (interviews and focus groups) with 30–35 service users and practitioners (from both alcohol and drug treatment services and existing stop smoking services). This will help us to identify factors that can act as barriers and enablers to service use and determine the core components of the intervention. For example: where the service should be based, what smoking cessation goals are realistic and relevant and what different types of smoking cessation support will be effective. This may include some or all of the following: one-to-one and/or group support delivered by an addiction worker trained in smoking cessation, free vaping kits, pharmacotherapy, a self-help smokefree phone app, https://smokefreeapp.com/, which offers smoking cessation support via a mobile phone.

The recruitment of participants will be from the client survey (respondents who completed the survey will also be offered the option to opt in to the qualitative research) and our existing networks and study advisory group. This is likely to include: policy makers from national and local government with a remit for drugs, alcohol or tobacco; third-sector organisations that address policy and/or provide services, care and support for people with a history of drug or alcohol problems; population health experts responsible for improving health and reducing inequalities by commissioning/managing tobacco control policy and services at local and national level; clinicians who care for people with drug and alcohol problems and provide or refer people to smoking cessation services; and academics researching drug, alcohol and tobacco problems.

#### 2.2.4. Expert Workshops

The final element of the development study will be two stakeholder workshops (each with around 25 delegates) where we will invite participants who contributed to the prior strands of the development study to take part in a 2-h workshop. We will present options for a draft service delivery plan and then engage in participatory activities (such as the Three Horizons Framework [30]) to elicit feedback and discussion to identify the strengths, weakness and elements for refinement and/or evaluation in the feasibility study. By the end of the development study, the service delivery protocol, logic model and TIDier checklist will be ready to test in the feasibility study.

### 2.3. Work Package 2: Feasibility Study

Work package 2 is a feasibility study in which key components of the proposed intervention will be tested (e.g., recruitment, provider adherence to agreed delivery protocol; client retention, acceptability) but without the need for randomisation and on a smaller scale than a pilot randomised control trial (RCT). The setting for the delivery of the feasibility study will be informed by findings from WP1 but is expected to be community service facilities including day centres, recovery hubs, specialist services premises and satellite clinics offering support for PDA across one Scottish NHS authority. The service will be delivered by an addiction counsellor based at the venue with smoking cessation expertise, involvement in the design and additional training in delivery of the model. Recruitment will, therefore, be a combination of service staff promoting the service to eligible individuals and the counsellor speaking directly to clients. Due to the additional vulnerabilities of this client group, we expect that the period of support will last around 16–20 weeks. Routine monitoring data will be collected: on referrals; attendance and retention; quit dates set; quit rates/reduction in cigarette intake at follow-up; type of behavioural support and NRT/vaping products use; and product cost. Smoking status will be measured using the Heaviness of Smoking Index (HSI) and carbon monoxide monitoring (which will give a reading of both smoking abstinence and reduction in tobacco consumption) at baseline and follow-ups [31].

#### Process Evaluation

Running alongside the feasibility study will be an embedded process evaluation that will assess the acceptability, feasibility and fidelity (via observation of service delivery) of the intervention. This will involve qualitative consultation (semi-structured, one to one/paired interviews) using purposive sampling, with service users (*n* = 15–20), service staff (*n* = 8–10) and the observation of service delivery. Data collected via the process evaluation will include: willingness of clients to engage with the service and of service providers to host the service and recruit participants into a future trial; time required for recruitment; feasibility of outcome measures. These data will inform the parameters of a future RCT including the required sample size (should results be positive) we will also explore what cost data can be collected and work out basic service delivery costs.

### 2.4. Analysis

Client survey data and results from the feasibility study will be entered into a database (IBM SPSS (version 28.01), IBM, Armonk, NY, USA) and analysed using descriptive frequencies and bivariate analysis. Interviews and focus groups will be recorded on a digital encrypted device and transcribed verbatim. We will use a thematic approach to analyse the data, facilitated by NVivo 12 (QSR international 2022, Burlington, MA, USA). First, we will read the transcripts to identify the key topics and issues that emerge from the data. Next, a draft analytical framework will be created, piloted, refined and finalised by the project team. Each transcript will then be coded and summarised into key themes using framework matrices or charts [32]. This approach reduces large volumes of data and facilitates a systematic approach to between-and-within case analysis. It also allows for emergent patterns and explanations to be explored and tested and thus provides the depth required for interpretative analysis, including enabling the features of future service models to emerge.

### 2.5. Ethics and Information Governance

The DASHES study was approved by the East Midlands–Leicester Central UK Research Ethics Committee on 1 September 2022 (22/EM/0167) and the study protocol is currently being registered with the ISRCTN registry (https://www.isrctn.com/).

All software will be hosted on a secure platform, and the use of NVivo 12 ensures that analysis is fully documented and conclusions can be clearly linked back to the original source data. Detailed information governance plans have been developed for the project, and all electronic files will be stored on secure university systems designed to hold individual level data with DASHES project files held in a secure project folder accessible only by named individuals on the research team. The project and project team will comply with University of Edinburgh and NHS Lothian standards regarding information governance and confidentiality.

## 3. Discussion

The outcome of this study will be a theoretically informed, co-created intervention with the potential to improve population health by supporting people with problematic drug or alcohol use to cut down or stop tobacco smoking. It will engage with an underserved group that is already marginalised, vulnerable to poor health and often hardly reached by smoking interventions and who will have experienced further inequality and disruption of support services due to the COVID-19 pandemic.

People on low incomes, with multimorbidity and from marginalised groups are less likely to be included in research, either as participants or advisers on design or ethics [33]. This project will go some way to addressing this. By developing a co-created service that builds on the evidence of how to reduce structural and relational barriers to access, engagement and retention in services, we will provide evidence that will be relevant to the provision of trauma-informed, inequality-sensitive services for other excluded and stigmatised groups. This highlights the potential contribution that the DASHES study could make to delivery of more inclusive, equitable patient care and service delivery. Further, by helping an underrepresented and stigmatised group stop smoking this will: improve their health and reduce the burden of smoking-related illness and treatment this population faces as well as the requirement for NHS care. In addition to health benefits, there may be financial benefits for participants whose smoking does reduce along with potential benefits for any peers or close family members, including children. For staff involved in service delivery, the study offers an opportunity to develop a greater awareness of and facilitate an attitudinal change towards smoking cessation and reduction, as well as to make links with other services. Findings will also be of interest to: people in recovery, commissioners, planners, managers, staff working in alcohol and drug services, NHS stop smoking services, GPs and other NHS staff, community-based staff in statutory and third-sector organisations who are working with people with a history of alcohol or substance issues; addiction researchers, public health staff and policy makers.

Findings will be used by the research team to apply for future funding for a subsequent pilot study prior to a definitive randomised control trial. Study findings will also inform future drug and alcohol policy and service provision for smoking cessation.

A particular strength of the DASHES study is its grounding in co-creation and bottom-up approaches to intervention design and testing. This research study was developed in response to a gap identified by addiction workers in the voluntary sector and NHS health professionals. As such, this study will be guided by a diverse collaboration of co-investigators from third-sector addiction and counselling services, NHS, people with lived experience of PDA and academics. A further strength of this study is the particular experience of the co-investigators with regards to the application of trauma-informed theory within the field of addiction work. This transferal of applied trauma-informed theory to a smoking cessation and reduction service will, therefore, be novel. However, these strengths could also be a limitation as the objectives for the study and its participatory approach are ambitious within the time scales and resources available.

## 4. Conclusions

Rates of tobacco smoking are significantly higher in groups with problematic drug or alcohol use than the general population, with no UK specialised stop smoking service for this client group. Existing services focus on successful quit attempts, which means that uptake among people with problematic drug or alcohol use is low, contributing to inequities in smoking cessation. This is in contrast with specialist substance misuse services, where any reduction in problematic drug and alcohol use by the client is seen as beneficial as part of a positive trajectory towards a more stable and better-quality life. Evidence shows that the pressure to achieve abstinence quickly (within the first two weeks of a standard programme) is difficult and demotivating for people who are in recovery from problematic substance use and can result in disengagement [10]. This study will develop and test a tailored, trauma-informed, harm reduction intervention that will support people recovering from problematic substance use to reduce or stop smoking. The intervention will be co-developed with service users, practitioners working in substance misuse services and researchers with experience designing harm reduction interventions.

## Figures and Tables

**Figure 1 ijerph-19-13709-f001:**
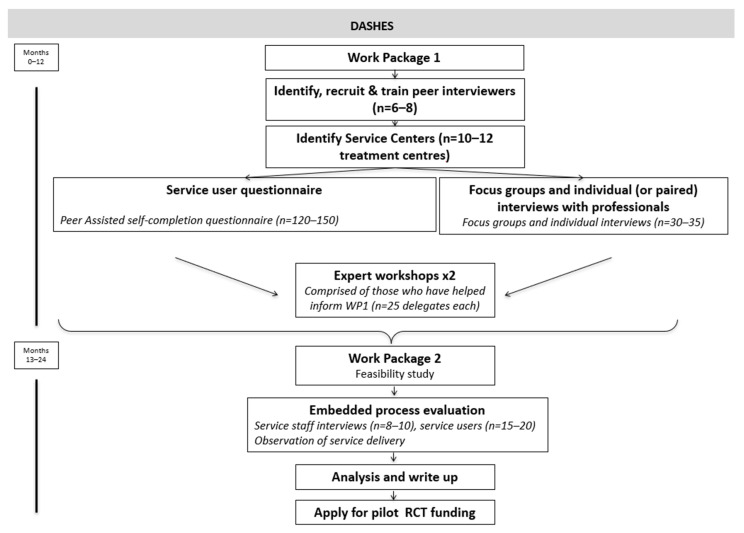
Overview of the study design.

## Data Availability

Not applicable.

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
