# Peer review of "DASHES Protocol: Development and Feasibility Testing of a Tailored Community Programme to Support People in Recovery from Problematic Alcohol and Drug Use to Cut Down or Stop Smoking Using Co-Creation"

_ijerph, 2022, doi:10.3390/ijerph192013709_

Round 1

Author Response

We are very grateful to reviewer 1 for their time and extremely helpful and constructive comments. Please find our response below. 

  1. Line 189- authors note they are recruiting individuals “at a stable point on their recovery journey.” How is this defined? Existing research among individuals with SUDs often defines stability differently for different substances and definitions vary.

We have added the following text to page 5, line 191 to clarify this point.

(defined as participants who have been accepted onto and are participating in a residential or community programme to address their problematic drug or alcohol use).

  1. Line 206- authors describe recruitment of participants for qualitative research from existing networks, study advisory groups and client surveys. Can the authors elaborate on the population for qualitative surveys and what populations are in their existing networks and advisory groups?

We have added the following text to page 5, line 206

Recruitment of participants will be from the client survey (respondents who completed the survey will also be offered the option to opt-in into the qualitative research) and our existing networks, study advisory group. This is likely to include: policy makers from National and Local Government with a remit for drugs, alcohol, or tobacco. Third sector organisations that address policy and/or provide services, care  and support for people with a history of drug or alcohol problems; population health experts responsible for improving health and reducing inequalities by commissioning/managing tobacco control policy and services at local and national level; clinicians that care for people with drug and alcohol problems and provide or refer people to smoking cessation services; and academics researching drug, alcohol and tobacco problems.

  1. Line 239-authors will assess fidelity of the intervention. How will this be assessed?

The following text has been added to page 6, line 250

Running alongside the feasibility study will be an embedded process evaluation that will assess acceptability, feasibility, and fidelity (via observation of service delivery) of the intervention

  1. Very minor typo on line 234, Heaviness of Smoking Index should be abbreviated HSI

Thank you this, has been amended.

I sincerely appreciate the opportunity to review this manuscript and this protocol paper and eventual study data will make a large contribution to the literature in addressing smoking in individuals with substance use.

Thank you very much for this encouragement and support. 

Reviewer 2 Report

This study proposed a protocol, to help those people with problematic drug and alcohol with smoking cessation. This study described details in framework and procedures step by step. Overall, this article is interesting and written well. I have a few concerns and comments as follows.

1.       I guess there is no need to provide abbreviation such as SSS and RCT in the abstract since they only appeared once.

2.       Page 2, line 61, SSS was first mentioned in here rather than in page 7, line 317.

3.       In background (page 2, line 63), authors mentioned the tobacco harm reduction approaches, could you please expand it more here? Are you referring to nicotine reduction or switching to other tobacco products such as e-cigarettes?

4.       Page 2, lines 84-87, please indicate the study design of using e-cigs as an effective smoking cessation tool (RCT?).

5.       I would suggest including more information regarding the day/community services. Why this method is potentially more effective, any evidence-based research on other aspects? It appeared like a sudden in the last paragraph of the background.

6.       For methods, how would authors define the PDA? How about the levels/degrees (moderate, mild, serve) of the PDA? By using DSM-V or other criteria?

7.       Page 6, line 25, I would refer a citation to NVivo 12 to help reader better understand the approach.

8.       For ethics, is this protocol/approach approved by IRB? Should mention it in the methods.

9.       Would it be feasible to provide the timeline of this project?

Author Response

We are very grateful to reviewer 2 for their time and extremely helpful and constructive comments. Please find our response below. 

This study proposed a protocol, to help those people with problematic drug and alcohol with smoking cessation. This study described details in framework and procedures step by step. Overall, this article is interesting and written well. I have a few concerns and comments as follows.

  1. I guess there is no need to provide abbreviation such as SSS and RCT in the abstract since they only appeared once.

Noted, this has been removed.

  1. Page 2, line 61, SSS was first mentioned in here rather than in page 7, line 317.

SSS text has been removed and lines 316 and 317 p7 have been edited to:

        or alcohol use than the general population, with no UK specialised stop smoking service  for this client group. Existing services focus on successful quit attempts

  1. In background (page 2, line 63), authors mentioned the tobacco harm reduction approaches, could you please expand it more here? Are you referring to nicotine reduction or switching to other tobacco products such as e-cigarettes?

Good point, it’s both. The following text has been added to clarify:

       without any harm reduction approaches being considered (e.g. cutting down or switching to less harmful products like e-cigarettes).

  1. Page 2, lines 84-87, please indicate the study design of using e-cigs as an effective smoking cessation tool (RCT?).

This has been clarified by the addition of the following text:

       Recent studies (using a pilot randomised controlled trial (RCT) and cross sectional study design) have found

  1. I would suggest including more information regarding the day/community services. Why this method is potentially more effective, any evidence-based research on other aspects? It appeared like a sudden in the last paragraph of the background.

      We respectfully disagree and feel the current text justifies our rationale for delivering the service in day centres only.

  1. For methods, how would authors define the PDA? How about the levels/degrees (moderate, mild, serve) of the PDA? By using DSM-V or other criteria?

      This is good point, but as participations for the study will already be receiving treatment for their PDA and we do not think using a validated tool is required.

  1. Page 6, line 25, I would refer a citation to NVivo 12 to help reader better understand the approach.

      This is already added.

  1. For ethics, is this protocol/approach approved by IRB? Should mention it in the methods.

      We assume IRB refer to the Institutional Review Board in the US, as noted in page 8 line 263-264 our study received ethical approval from the UK. We have edited the text to make this clearer. It now reads:

       The DASHES study was approved by the East Midlands - Leicester Central UK Research Ethics Committee on the 1st September 2022 (22/EM/0167)

  1. Would it be feasible to provide the timeline of this project?

The following text has been added to page 3, line 133,134

       The DASHES study design is comprised of two work packages, delivered over a 24 month period, summarised in Figure 1.